# Cross-reactive antibodies after SARS-CoV-2 infection and vaccination

Marloes Grobben[1†], Karlijn van der Straten[1,2†], Philip JM Brouwer[1], Mitch Brinkkemper[1], Pauline Maisonnasse[3], Nathalie Dereuddre-Bosquet[3], Brent Appelman[4], AH Ayesha Lavell[5], Lonneke A van Vught[4], Judith A Burger[1], Meliawati Poniman[1], Melissa Oomen[1], Dirk Eggink[6], Tom PL Bijl[1], Hugo DG van Willigen[1], Elke Wynberg[2,7], Bas J Verkaik[1], Orlane JA Figaroa[1], Peter J de Vries[8], Tessel M Boertien[8], Amsterdam UMC COVID-19 S3/HCW study group, Marije K Bomers[5], Jonne J Sikkens[5], Roger Le Grand[3], Menno D de Jong[1], Maria Prins[2,7], Amy W Chung[9], Godelieve J de Bree[2], Rogier W Sanders[1,10], Marit J van Gils[1]*

[1]Department of Medical Microbiology, Amsterdam UMC, University of Amsterdam, Amsterdam Institute for Infection and Immunity, Amsterdam, Netherlands; [2]Department of Internal Medicine, Amsterdam UMC, University of Amsterdam, Amsterdam Institute for Infection and Immunity, Amsterdam, Netherlands; [3]Center for Immunology of Viral, Auto-immune, Hematological and Bacterial Diseases (IMVA-HB/IDMIT), Université Paris-Saclay, INSERM, CEA, Fontenay-aux-Roses, France; [4]Center for Experimental and Molecular Medicine, Amsterdam UMC, University of Amsterdam, Amsterdam Institute for Infection and Immunity, Amsterdam, Netherlands; [5]Department of Internal Medicine, Amsterdam UMC, Vrije Universiteit Amsterdam, Amsterdam Institute for Infection and Immunity, Amsterdam, Netherlands; [6]National Institute for Public Health and the Environment, RIVM, Bilthoven, Netherlands; [7]Department of Infectious Diseases, Public Health Service of Amsterdam, GGD, Amsterdam, Netherlands; [8]Department of Internal Medicine, Tergooi Hospital, Amsterdam, Netherlands; [9]Department of Microbiology and Immunology, Peter Doherty Institute for Infection and Immunity, The University of Melbourne, Victoria, Australia; [10]Department of Microbiology and Immunology, Weill Medical College of Cornell University, New York, United States

*For correspondence:
m.j.vangils@amsterdamumc.nl

[†]These authors contributed equally to this work

Group author details:
Amsterdam UMC COVID-19 S3/HCW study group See page 15

**Competing interest:** The authors declare that no competing interests exist.

**Abstract** Current SARS-CoV-2 vaccines are losing efficacy against emerging variants and may not protect against future novel coronavirus outbreaks, emphasizing the need for more broadly protective vaccines. To inform the development of a pan-coronavirus vaccine, we investigated the presence and specificity of cross-reactive antibodies against the spike (S) proteins of human coronaviruses (hCoV) after SARS-CoV-2 infection and vaccination. We found an 11- to 123-fold increase in antibodies binding to SARS-CoV and MERS-CoV as well as a 2- to 4-fold difference in antibodies binding to seasonal hCoVs in COVID-19 convalescent sera compared to pre-pandemic healthy donors, with the S2 subdomain of the S protein being the main target for cross-reactivity. In addition, we detected cross-reactive antibodies to all hCoV S proteins after SARS-CoV-2 vaccination in macaques and humans, with higher responses for hCoV more closely related to SARS-CoV-2. These findings support the feasibility of and provide guidance for development of a pan-coronavirus vaccine.

## Editor's evaluation

To develop a pan-coronavirus vaccine, the authors investigated the presence and specificity of cross-reactive antibodies against the spike proteins of human coronaviruses after SARS-CoV-2 infection and vaccination and quantified cross-reactive antibodies.

## Introduction

One and a half year after the emergence of severe acute respiratory coronavirus 2 (SARS-CoV-2), the causative agent of coronavirus infectious disease 2019 (COVID-19), the pandemic is still a major global issue with unprecedented consequences on healthcare systems and economies. Following the rapid initiation of many COVID-19 vaccine studies, the first vaccines are already FDA/EMA approved and mass vaccination campaigns are rolled out to hopefully soon subdue this pandemic (*Polack et al., 2020*; *Baden et al., 2021*; *Voysey et al., 2021*; *Sadoff et al., 2021*). However, these vaccines show reduced efficacy against emerging SARS-CoV-2 variants and vaccine breakthrough infections are frequently reported (*Madhi et al., 2021*; *Lopez Bernal et al., 2021*; *Kustin et al., 2021*; *Hacisuleyman et al., 2021*). To anticipate emerging variants, more broadly protective SARS-CoV-2 vaccines are desirable. However, the ultimate goal should be a pan-coronavirus vaccine that would be able to induce broad immunity and protection against multiple coronaviruses, thereby preparing us for future outbreaks.

New coronaviruses that infect humans (hCoVs) emerge frequently, and SARS-CoV-2 is the fifth to be discovered in less than two decades, bringing the total up to seven (*Kahn and McIntosh, 2005*). These include the seasonal betacoronaviruses hCoV-OC43 and hCoV-HKU1 as well as alphacoronaviruses hCoV-NL63 and hCoV-229E, which usually cause mild respiratory symptoms and account for approximately 5–30% of common colds during the winter season (*Zhu et al., 2020*; *Li et al., 2020*). In contrast, severe acute respiratory syndrome (SARS)-CoV and Middle Eastern Respiratory Syndrome (MERS)-CoV are far more pathogenic and led to epidemics in 2003 and 2013–2015, respectively (*da Costa et al., 2020*; *de Wit et al., 2016*). The currently circulating SARS-CoV-2 is less pathogenic compared to SARS-CoV and MERS-CoV, resulting in mild flu-like symptoms in the majority of patients, while only a small group of patients develop (bilateral) pneumonia that can rapidly deteriorate in severe acute respiratory distress syndrome (*Tabata et al., 2020*). Despite the lower pathogenicity compared to SARS-CoV and MERS-CoV, the high infection rate of SARS-CoV-2 results in large numbers of hospitalizations and deaths worldwide.

The majority of SARS-CoV-2 vaccines are based on the spike (S) glycoprotein, a homotrimeric glycoprotein that is present on the surface of all coronaviruses and is the main target of protective antibodies. The S protein plays a pivotal role in viral entry and consists of an S1 subdomain including the receptor-binding domain (RBD) and an S2 subdomain containing the fusion peptide (*Wrapp et al., 2020*; *Walls et al., 2020*). The S protein of SARS-CoV-2 is most closely related to the S protein of SARS-CoV (~75% amino acid sequence identity), shows less similarity to MERS-CoV S protein (~50% identity), and is more distinct from the S proteins of the seasonal hCoVs (~25–30% identity) (*Lu et al., 2020*; *Barnes et al., 2020*). Although this sequence identity is considered low from the perspective of eliciting cross-reactive antibodies by vaccination, several studies observed cross-reactivity of COVID-19 sera against other hCoVs (*Westerhuis et al., 2020*; *Dugas et al., 2021*; *Ng et al., 2020*; *Aydillo et al., 2021*; *Song et al., 2021*). In addition, the isolation and engineering of broader hCoV monoclonal antibodies have been reported (*Wec et al., 2020*; *Rappazzo et al., 2021*; *Wang et al., 2020*).

In this study, we identify broad hCoV antibody responses following SARS-CoV-2 infection and vaccination. We identify the S2 subdomain as a prominent target for cross-reactive antibodies and demonstrate that cynomolgus macaques vaccinated with a SARS-CoV-2 S protein nanoparticle vaccine elicit detectable antibody responses to other hCoVs. These results will guide the development of a pan-coronavirus vaccine.

## Results

### Convalescent COVID-19 sera contain robust IgG responses to the SARS-CoV-2 S protein

Sera from 50 PCR-confirmed SARS-CoV-2-infected patients, aged 22–75 years, who suffered from a range of COVID-19 disease severities were obtained 4–6 weeks after symptom onset (*Supplementary file 1*). A custom Luminex assay was used to measure IgG antibody levels to the prefusion-stabilized trimeric SARS-CoV-2 spike (S) protein in these convalescent COVID-19 sera and compared to pre-pandemic healthy donor sera.

Sera from convalescent COVID-19 patients showed 990-fold higher antibody levels to SARS-CoV-2 S protein compared to pre-pandemic healthy donor sera (median 3.8 $\log_{10}$ vs. 0.8 $\log_{10}$ median fluorescent intensity [MFI], p<0.0001) (*Figure 1a*). This is in line with a positive serological response in all but one of the participants using the commercially available WANTAI SARS-CoV-2 RBD total immunoglobulin serum ELISA (*Figure 1a* and *Supplementary file 1*). When subdividing the convalescent patients based on admission status, fourfold higher antibody levels were observed for hospitalized patients compared to nonhospitalized patients (median 4.1 $\log_{10}$ vs. 3.5 $\log_{10}$ MFI, p<0.0001). SARS-CoV-2 neutralization capacity was also significantly higher (fivefold) in hospitalized patients (3.6 $\log_{10}$ vs. 2.9 $\log_{10}$ MFI, p<0.0001), and antibody levels and neutralization capacity were strongly correlated in all patients (Spearman's $r$ = 0.735, p<0.0001) (*Figure 1b*). An eightfold higher antibody response to the SARS-CoV-2 S protein was observed in the oldest age group of COVID-19 patients (age >60) compared to the youngest age group (age <35) (4.1 $\log_{10}$ vs. 3.2 $\log_{10}$ MFI, p<0.001), which is in line with age being a risk factor for severe COVID-19 (*Figure 1a*; *Williamson et al., 2020*). No difference in antibody levels was observed between sex (3.8 $\log_{10}$ vs. 3.9 $\log_{10}$ MFI, p=0.71). These findings were confirmed with a principal component analysis, which showed two distinct groups based on hospital admission status, and indicated age as an important contributing variable (*Figure 1c*, *Figure 1—figure supplement 1*). Overall, these results are consistent with previous studies (*Ju et al., 2020*; *Lynch et al., 2021*; *Yang et al., 2021*).

### Convalescent COVID-19 sera are cross-reactive with all hCoV S proteins

Next, we measured IgG binding to the S proteins of other hCoVs: SARS-CoV, MERS-CoV, hCoV-OC43, hCoV-HKU1, hCoV-229E, and hCoV-NL63. We found significantly higher levels of antibodies binding to all hCoV S proteins in convalescent COVID-19 sera compared to pre-pandemic healthy donor sera, with the largest difference for SARS-CoV and MERS-CoV S protein (123- and 11-fold difference between medians, respectively, p<0.0001 for both proteins) (*Figure 2a*). For the circulating seasonal hCoVs, a two- to fourfold difference in median levels of antibodies binding to the S proteins was observed; hCoV-OC43 (4.2-fold, p<0.0001), hCoV-HKU1 (4.3-fold, p<0.0001), hCoV-229E (3.8-fold, p<0.0001), and hCoV-NL63 (1.8-fold, p<0.01). No such difference was found for the control protein tetanus toxoid (0.8-fold, p=0.85) (*Figure 2—figure supplement 1a*).

To further explore the specificity of the cross-reactive antibodies in convalescent COVID-19 sera, we performed a depletion with soluble SARS-CoV-2 S protein (*Figure 2b*, *Figure 3—figure supplement 1*). Depletion of convalescent COVID-19 sera resulted in a 92% reduction of antibody levels against the SARS-CoV-2 S protein. The reduction of antibodies binding to the other hCoV S proteins was strongest for SARS-CoV (61% mean reduction) and MERS-CoV (25%). A more modest reduction of signal was observed for hCoV-OC43 (13% mean reduction) and hCoV-229E S protein (6%), and we observed no depletion of antibodies binding to hCoV-HKU1 (2%) and hCoV-NL63 (–2%). On the individual level, some patients showed up to 72, 70, and 34% reduction of antibodies binding to hCoV-OC43, hCoV-229E and hCoV-HKU1 S protein after depletion, respectively. No reduction in signal was observed for the control protein tetanus toxoid after SARS-CoV-2 S depletion (*Figure 2b*, *Figure 2—figure supplement 1b*). Antibodies binding to other hCoV S proteins in pre-pandemic healthy donor sera did not decrease after depletion (*Figure 2—figure supplement 1c*). These depletion experiments corroborate the finding that convalescent COVID-19 sera contain cross-reactive antibodies able to recognize other hCoV S proteins. We observed the highest cross-reactivity for the more closely related hCoVs (*Supplementary file 2*). Indeed, the level of cross-reactivity expressed as the reduction of binding antibodies after depletion by SARS-CoV-2 S protein was associated with the level of sequence identity (Spearman's $r$ = 0.83, p=0.06) (*Figure 2—figure supplement 2*).

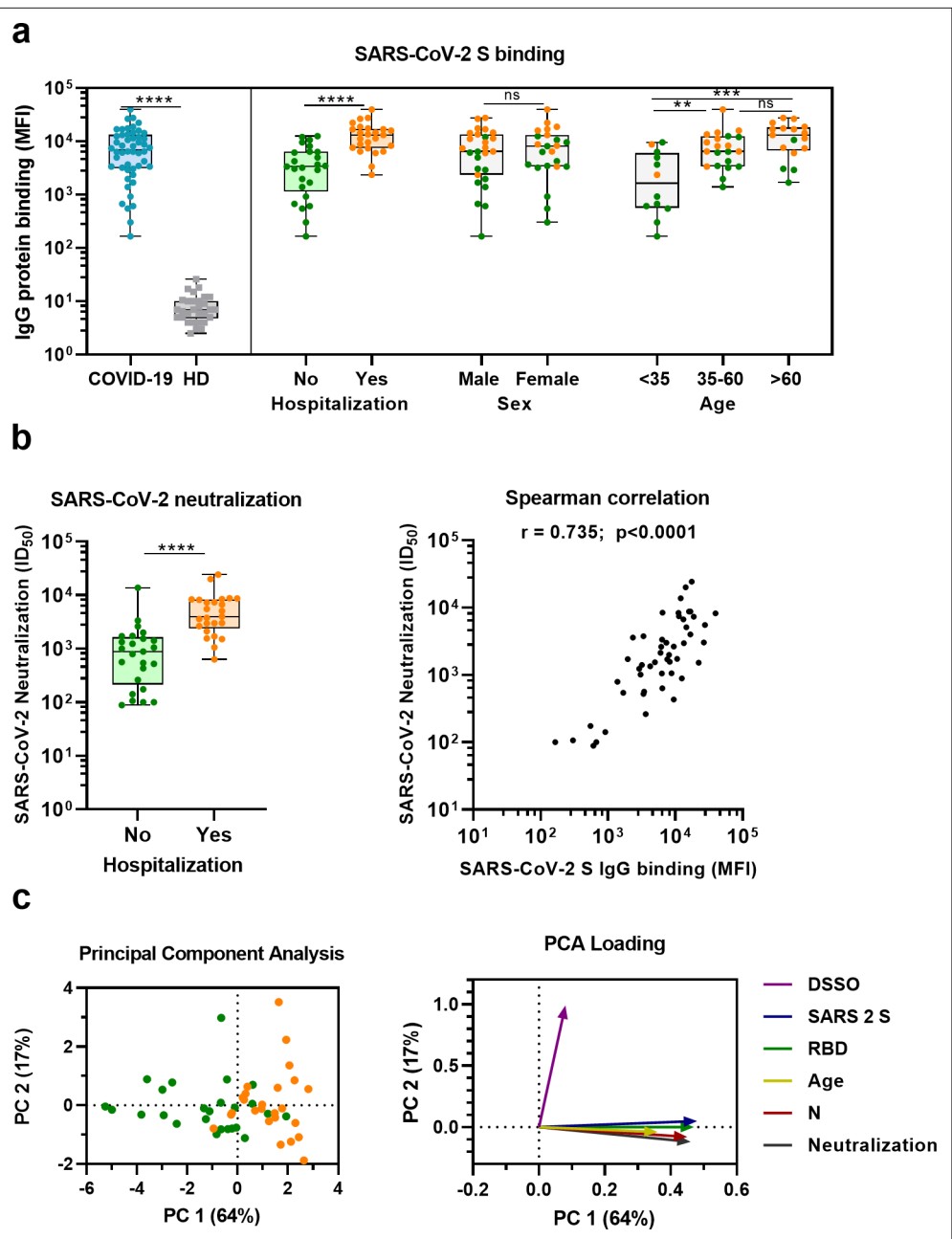

**Figure 1.** SARS-CoV-2 antibody response in COVID-19 patients. (**a**) IgG binding of sera to SARS-CoV-2 S protein measured with a custom Luminex assay. Convalescent COVID-19 sera (COVID-19, blue dots, n = 50) were compared to sera of pre-pandemic healthy donors (HD, gray squares, n = 30) using a Mann–Whitney U test. The cohort of COVID-19 patients was subdivided according to admission status, sex, and age. Admission status is indicated with orange dots for hospitalized patients (Yes) and green dots if no admission to hospital was needed (No). (**b**) SARS-CoV-2 pseudovirus neutralization capacity compared between hospitalized patients (orange dots) and patients that did not need admission to hospital (green dots) (left). The correlation between the SARS-CoV-2 pseudovirus neutralization and SARS-CoV-2 S antibody levels in the Luminex assay was determined using Spearman's correlation (right). (**c**) A principal component analysis (PCA) was performed using the following variables: IgG antibodies binding to SARS-CoV-2 S protein (S), RBD and nucleocapsid protein (N), SARS-CoV-2 neutralization, age and days since symptom onset (DSSO). Admission status is indicated with orange dots for hospitalized patients and green dots if no admission to hospital was needed (left). Loading of the principal component (PC) plot by each variable is indicated with colored arrows to visualize the contribution of each variable on PC 1 and 2 (right). ns, not significant; **p<0.01; ***p<0.001; ****p<0.0001. MFI, median fluorescent intensity; S, spike protein; ID$_{50}$, serum dilution at which 50% of pseudovirus is neutralized; r, Spearman's rank correlation

*Figure 1 continued on next page*

*Figure 1 continued*

coefficient.

The online version of this article includes the following figure supplement(s) for figure 1:

**Source data 1.** Source data of all panels of *Figure 1*.

**Figure supplement 1.** Principal component analysis showing the influence of sex and disease severity on the SARS-CoV-2 IgG response.

---

To further explore the possible functionality of these cross-reactive antibodies, we investigated the correlation between hCoV S protein binding antibodies and SARS-CoV-2 neutralization titers. In addition to the observed correlation between SARS-CoV-2 S protein antibody levels and SARS-CoV-2 neutralization, we found a moderate correlation between SARS-CoV-2 neutralization and antibodies binding to SARS-CoV S protein ($r = 0.545$, $p<0.0001$), a weak correlation with antibodies that bind to MERS-CoV, hCoV-HKU1, and hCoV-229E S protein ($r = 0.324$, 0.308, 0.320, respectively, all $p<0.05$), and no correlation with antibodies binding to hCoV-OC43 and hCoV-NL63 S protein ($r = 0.188$ and $-0.095$, $p>0.05$) (*Figure 2—figure supplement 3*). These correlations suggest that cross-reactive antibodies form a minor component of the neutralizing antibodies induced by SARS-CoV-2 infection. We also investigated if there was an association between cross-reactivity and disease severity with a principal component analysis. Antibodies binding to other hCoV S proteins did not contribute to the separation between patients based on admission status as clearly as was seen for SARS-CoV-2 S protein antibody levels, suggesting a limited impact of cross-reactive antibodies on disease severity (*Figure 2—figure supplement 4*).

## Cross-reactive antibodies preferentially target the S2 subdomain

To determine the main target on the SARS-CoV-2 S protein for cross-reactive antibodies, we performed a depletion experiment in which we compared the reduction of IgG binding to hCoV S proteins after depleting the sera with soluble monomeric SARS-CoV-2 S1 or S2 subdomains. After depletion with the SARS-CoV-2 S2 subdomain, a 48, 20, and 19% mean reduction of signal was observed for antibodies binding to SARS-CoV, MERS-CoV, and hCoV-OC43 S protein, respectively, while no reduction was observed after S1 depletion (*Figure 3*). Absolute MFI values before and after depletion are shown in *Figure 3—figure supplement 2a*. For some individuals, depletion rates reached up to 83% for SARS-CoV, 75% for MERS-CoV, and 80% for hCoV-OC43 S protein while there were no individuals with substantial reduction of cross-reactive antibodies after S1 depletion. We did not observe significant depletion of antibodies binding to hCoV-HKU1 and hCoV-229E S proteins by S1 or S2 subdomains of SARS-CoV-2 at the group level. However, in some individuals the signal was reduced by up to 59% for hCoV-HKU1 and 71% for hCoV-229E S protein following depletion with the SARS-CoV-2 S2 subdomain, while such effects were not observed after S1 subdomain depletion. Changes in antibodies that bind to hCoV-NL63 S protein after depletion with SARS-CoV-2 S1 and S2 subdomains were comparable to those observed with the control protein tetanus toxoid and to the reduction in signal for hCoV S in healthy donors (*Figure 3—figure supplement 1b* and *Figure 3—figure supplement 2b*). SARS-CoV-2 S2 subdomain depletion was significantly correlated with S2 sequence identity ($r = 0.90$, $p=0.03$) while there was no correlation for SARS-CoV-2 S1 subdomain depletion ($r = 0.23$, $p=0.67$) (*Figure 2—figure supplement 2*). These results indicate that S2-specific antibodies have a greater contribution to cross-reactivity than S1-specific antibodies, which is in accordance with the higher conservation of the S2 subdomain compared to the S1 subdomain (*Supplementary file 2*; *Nguyen-Contant et al., 2020*).

## SARS-CoV-2 S protein vaccination elicits hCoV cross-reactive antibodies in macaques

The observed cross-reactive antibody response following natural infection with SARS-CoV-2 does not reveal whether these might be de novo responses or the result of cross-boosted preexisting memory from previous hCoV infections. To investigate the ability of SARS-CoV-2 vaccination to induce de novo cross-reactive antibodies against hCoVs, we investigated the serum of six cynomolgus macaques that were vaccinated three times with prefusion-stabilized trimeric SARS-CoV-2 S protein coupled to I53-50 nanoparticles (*Brouwer et al., 2021*). The vaccination response was studied by comparing serum IgG

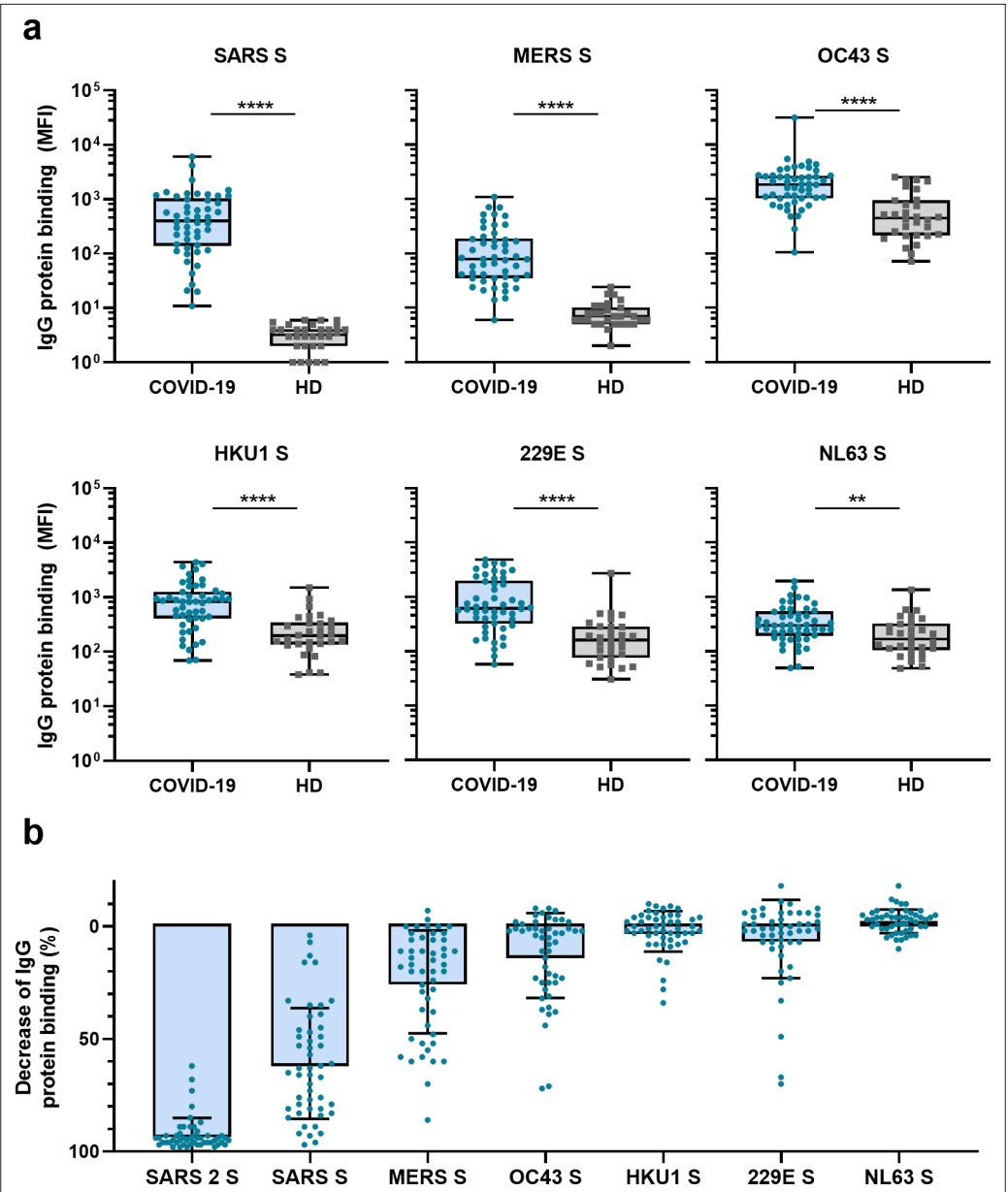

**Figure 2.** Cross-reactivity to hCoV S proteins in convalescent COVID-19 sera. (**a**) IgG binding to all hCoV S proteins measured with a custom Luminex assay in convalescent COVID-19 sera (COVID-19, blue dots, n = 50) were compared to pre-pandemic sera from healthy donors (HD, gray squares, n = 30) using a Mann–Whitney U test. Box plots range the minimum and maximum values. **p<0.01; ****p<0.0001. Serum IgG binding to the tetanus toxoid control protein is shown in *Supplementary file 2a*. (**b**) Percent decrease of IgG binding to all other hCoV S proteins in COVID-19 patient sera (n = 50) after depletion with soluble recombinant SARS-CoV-2 S protein. Bars represent the mean decrease of binding IgG as a percentage of the total binding IgG observed in undepleted sera, and error bars represent the standard deviation. Dots represent the percent decrease of binding IgG observed in individual sera. The percent decrease of IgG binding to tetanus toxoid control protein in patient sera and healthy donor sera after SARS-CoV-2 S protein depletion is shown in *Figure 2—figure supplement 1b and c*, respectively, and depletion of hCoV S IgG binding in healthy donor sera is shown in *Figure 2—figure supplement 1d*. All absolute median fluorescent intensity (MFI) values are shown in *Figure 3—figure supplement 1a* and *Figure 3—figure supplement 2a*. S, spike protein.

The online version of this article includes the following source data and figure supplement(s) for figure 2:

**Source data 1.** Source data of all panels of *Figure 2*.

**Figure supplement 1.** Antibody reactivity to tetanus toxoid in convalescent COVID-19 sera and depletion of

*Figure 2 continued on next page*

*Figure 2 continued*

antibodies binding to all proteins in healthy donors.

**Figure supplement 2.** Correlation between sequence identity and reduction of cross-reactive antibodies in depletion assay for S, S1, and S2.

**Figure supplement 3.** Spearman's correlations between SARS-CoV-2 neutralization and antibody reactivity to hCoV S and tetanus toxoid.

**Figure supplement 4.** Principal component analysis including SARS-CoV-2 binding antibodies, neutralization, clinical characteristics, and cross-reactivity.

**Figure supplement 5.** Native PAGE analysis of hCoV proteins.

**Figure supplement 5—source data 1.** Raw, unedited, and uncropped pictures of colloidal blue stainings of 4–12% NuPAGE Bis-Tris gels, and uncropped pictures with the sizes of the marker in kilo Dalton and the relevant bands indicated.

**Figure supplement 6.** Reproducibility of the Luminex assay.

binding to hCoV S proteins after three vaccinations (week 12) to the pre-immunization baseline (week 0). We observed that all macaques developed a detectable increase of antibodies that bind to all hCoV S proteins. Antibodies binding to SARS-CoV, MERS-CoV, and hCoV-OC43 S proteins were increased compared to baseline by 3313-, 168-, and 106-fold, respectively (*Figure 4a*). This cross-reactivity was already detectable 2 weeks after the first immunization (*Figure 4—figure supplement 1a*). The induction of antibodies that bind to hCoV-HKU1 and hCoV-229E S proteins was relatively low with a 39- and 16-fold increase from baseline, respectively. Antibodies binding to hCoV-NL63 S protein showed only a sixfold increase from baseline (*Figure 4a*), and there was no increase observed for tetanus toxoid (*Figure 4—figure supplement 1b and c*). All comparisons were statistically significant except for the tetanus toxoid. Using the SARS-CoV-2 S protein depletion assay, an 88, 98, and 99% mean reduction of antibodies binding to SARS-CoV, MERS-CoV, and hCoV-OC43 S protein, respectively, was observed (*Figure 4b*). In addition, depletion also resulted in a mean reduction of antibodies binding to proteins hCoV-229E S (92%), hCoV-HKU1 S (92%), and hCoV-NL63 S (75%). Virtually no depletion (3%) was observed for the control protein tetanus toxoid (*Figure 4—figure supplement 2*). Together, these

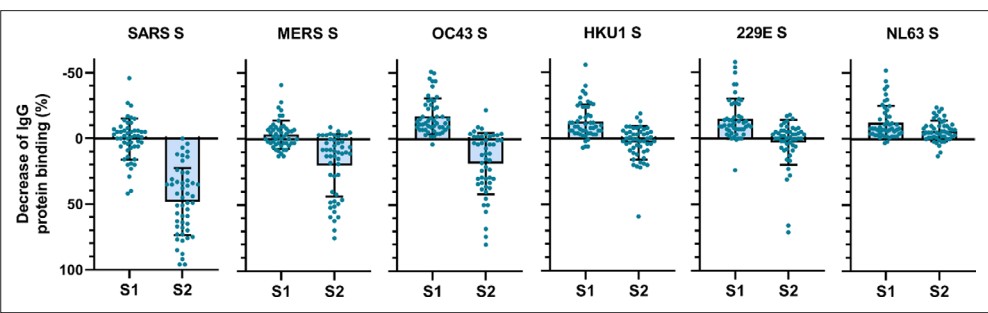

**Figure 3.** Depletion of S1 and S2 subdomain-specific cross-reactivity in convalescent COVID-19 sera. Percent decrease of IgG binding to all other hCoV S proteins in convalescent COVID-19 sera (n = 50) after depletion with soluble recombinant monomeric SARS-CoV-2 S1 or S2 subdomains. Bars represent the mean decrease of binding IgG as a percentage of the total binding IgG observed in undepleted sera, and error bars represent the standard deviation. Dots represent the percent decrease of binding IgG observed in individual sera. Percent decrease of IgG binding to tetanus toxoid control protein in patient sera after SARS-CoV-2 S1 and S2 depletion is shown in *Figure 3—figure supplement 1a*. The percent decrease of IgG binding to hCoV S and tetanus toxoid protein in healthy donor sera after SARS-CoV-2 S1 and S2 depletion is shown in *Figure 3—figure supplement 1b and c*. All absolute median fluorescent intensity (MFI) values are shown in *Figure 3—figure supplement 1a* and *Figure 3— figure supplement 2a*. S, spike protein.

The online version of this article includes the following figure supplement(s) for figure 3:

**Source data 1.** Source data of all panels of *Figure 3*.

**Figure supplement 1.** Raw data and control data from depletion assays on convalescent COVID-19 sera.

**Figure supplement 2.** Raw data and control data from depletion assays on healthy donor sera.

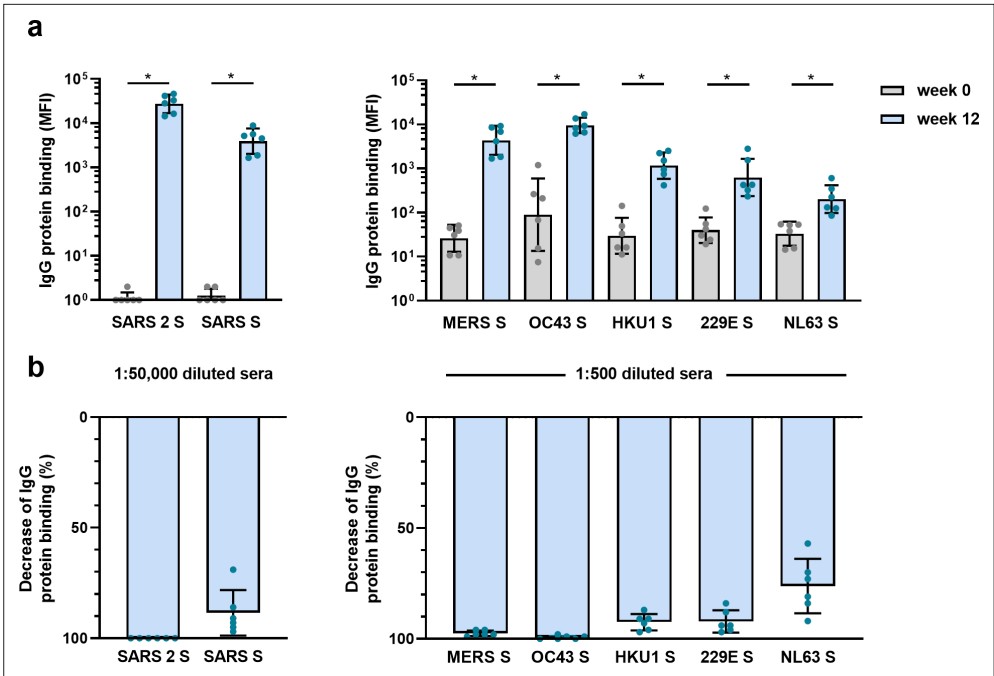

**Figure 4.** Cross-reactivity and depletion of cross-reactivity to hCoV S proteins in immunized macaques. (**a**) SARS-CoV-2 S protein-specific IgG binding and cross-reactive IgG binding to SARS-CoV S protein at week 0 (pre-immunization baseline) and week 12 (after a total of three immunizations), measured with a custom Luminex assay in 1:50,000 diluted serum of six cynomolgus macaques immunized with a SARS-CoV-2 S nanoparticle vaccine (left) and cross-reactive IgG binding to all other hCoV S proteins, measured in 1:500 diluted serum in the same animals (right). Bars represent the geometric mean of the six animals, and error bars the geometric standard deviation; dots represent the individual animals, statistical comparisons used a Wilcoxon matched-pairs signed rank test. IgG binding to hCoV S and tetanus toxoid control protein at weeks 2–12 is shown in *Figure 4—figure supplement 1a and b*, and IgG binding to tetanus toxoid control protein at weeks 0 and 12 is shown in *Figure 4—figure supplement 1c*. *p<0.05. (**b**) Percent decrease of IgG binding to all hCoV S proteins after depletion with soluble recombinant SARS-CoV-2 S protein. Bars represent the mean decrease of binding IgG as a percentage of the total binding IgG observed in undepleted sera of the six cynomolgus macaques at week 12, and error bars represent the standard deviation. Dots represent the percent decrease of binding IgG observed in individual sera. The percent decrease of IgG binding to tetanus toxoid control protein is shown in *Figure 4—figure supplement 2b*, and all median fluorescent intensity (MFI) values are shown in *Figure 4—figure supplement 2a*. S, spike protein.

The online version of this article includes the following figure supplement(s) for figure 4:

**Source data 1.** Source data of all panels of *Figure 4*.

**Figure supplement 1.** Antibodies binding to hCoV S and tetanus toxoid in immunized cynomolgus macaques over time.

**Figure supplement 2.** Reactivity and depletion of antibodies to hCoV S proteins and tetanus toxoid in immunized cynomolgus macaques.

results demonstrate that the SARS-CoV-2 S protein is capable of inducing cross-reactive antibodies to both alpha and beta hCoVs in macaques.

## COVID-19 mRNA vaccine elicits hCoV cross-reactive antibodies in SARS-CoV-2-naïve individuals

To confirm the ability of SARS-CoV-2 vaccination to induce cross-reactive antibodies also in a human setting, and to investigate the induction of these antibodies elicited by vaccination in a background of preexisting hCoV immunity, we measured IgG binding to hCoVs S proteins after vaccination in 45 SARS-CoV-2-naïve individuals. Antibody levels 3 weeks after the first vaccination and 4 weeks after the second vaccination with the Pfizer-BioNTech mRNA vaccine were compared with their baseline results before vaccination (*Figure 5*). IgG binding to hCoV S proteins is already increased after the first vaccination, but the largest increase in binding was observed following the second vaccination

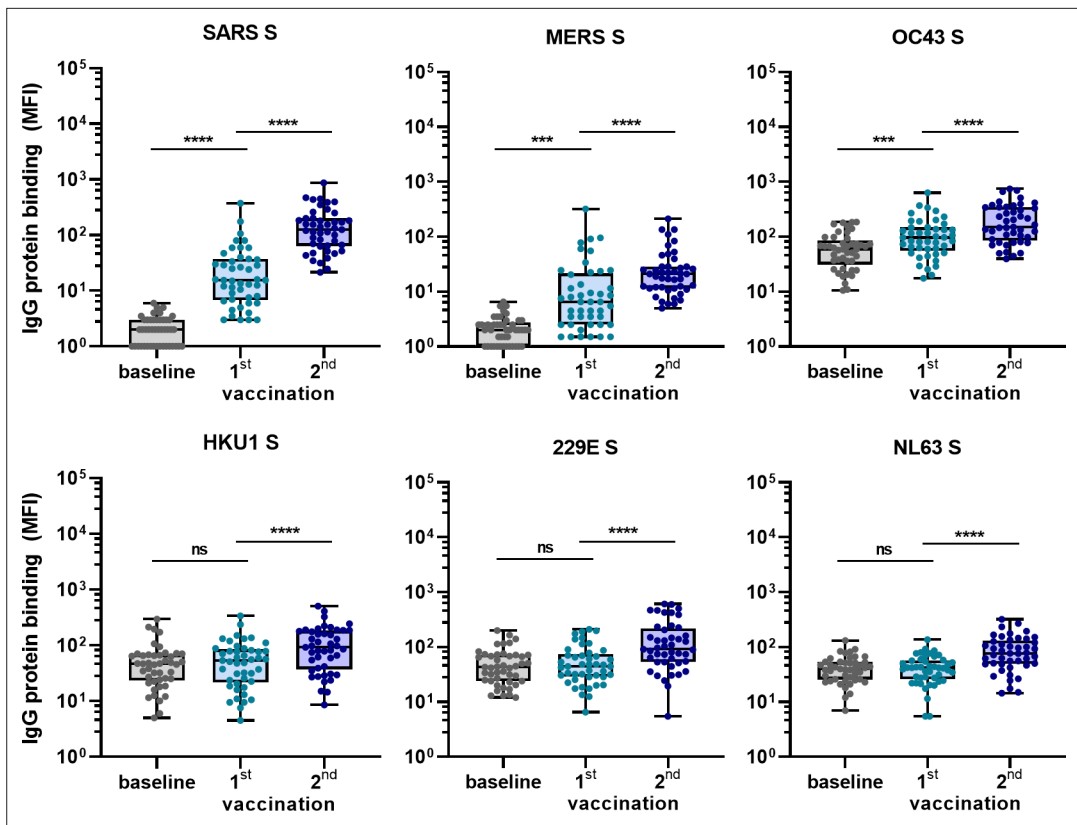

**Figure 5.** Cross-reactivity to hCoV S proteins following COVID-19 mRNA vaccination in human IgG binding to hCoVs S proteins measured with a custom Luminex assay in 1:100,000 diluted serum of 45 SARS-CoV-2-naïve individuals at baseline (gray dots), 3 weeks after their first vaccination (blue dots) and 4 weeks after their second vaccination (dark blue dots) with Pfizer-BioNTech mRNA vaccine. IgG binding to tetanus toxoid control protein is shown in *Figure 5—figure supplement 1*. Friedman test with Dunn's multiple comparisons test was used to compare medians of different time points and correct for multiple testing. Box plots range the minimum and maximum values. ns, not significant; ***$p<0.001$; ****$p<0.0001$.

The online version of this article includes the following figure supplement(s) for figure 5:

**Source data 1.** Source data of all panels of *Figure 5*.

**Figure supplement 1.** Antibodies binding to tetanus toxoid protein following Pfizer-BioNTech mRNA vaccination in humans.

for SARS-CoV, MERS-CoV, and hCoV-OC43 (62-fold, 11-fold, and 2.5-fold, respectively, all $p<0.0001$) (*Figure 5*). This is in accordance with our previous results in SARS-CoV-2-infected individuals and immunized macaques. The increase observed against hCoV-HKU1 (2.0-fold), hCoV-229E (2.1-fold), and hCoV-NL63 (1.9-fold) was similar to the increase observed for the control protein tetanus toxoid (1.7-fold, $p<0.0001$) (*Figure 5—figure supplement 1*). These results confirm our previous findings that most cross-reactivity following SARS-CoV-2 vaccination is observed for the most closely related hCoVs.

## Discussion

As mass vaccination campaigns take place around the globe, there is optimism that the SARS-CoV-2 pandemic will be subdued. However, there will likely be future challenges with emerging SARS-CoV-2 variants as well as with entirely new hCoVs. A number of studies have shown that neutralizing antibodies induced by COVID-19 vaccines have substantially reduced activity against some SARS-CoV-2 variants (*Voysey et al., 2021*; *Wu et al., 2021*; *Liu et al., 2021*; *Xie et al., 2021*). Therefore, development of broader SARS-CoV-2 vaccines is needed, with a pan-coronavirus vaccine as the ultimate goal.

In this study, we show that convalescent COVID-19 sera have higher levels of antibodies that bind to other hCoV S proteins compared to pre-pandemic healthy donors. We confirmed this cross-reactivity by showing that SARS-CoV-2 S protein could partially deplete antibodies binding to all hCoV S proteins except hCoV-NL63 in COVID-19 patient sera, while no effect of SARS-CoV-2 S protein depletion was observed in healthy donor sera. Additionally, we see that the elevation of hCoV S antibody levels and the capacity of SARS-CoV-2 S protein to deplete these antibodies is highest for hCoV S proteins that share most sequence homology with the SARS-CoV-2 S protein. Cross-reactive antibodies can be the result of de novo responses or due to cross-boosting of preexisting hCoV immunity. Others have shown cross-reactive antibodies resulting from boosting of preexisting immunity after natural infection in COVID-19 patients (*Ng et al., 2020*; *Anderson et al., 2021*). We further explored the possibility to induce de novo cross-reactive antibodies by vaccination in a cynomolgus macaque model. These macaques are unlikely to have encountered hCoVs, although infections of macaques have been described for the common cold hCoVs (*Dijkman et al., 2009*) and the animals were not tested for hCoV infection prior to entering the study. All SARS-CoV-2 S protein nanoparticle immunized macaques showed detectable antibodies binding to all hCoV S proteins, which could also be depleted with SARS-CoV-2 S protein. We suspect that the fact that not all sera reached a 100% depletion could be an indication of conformational differences between our hCoV S proteins, even though we confirmed that they form trimers (*Figure 2—figure supplement 5*). Throughout this study, a tetanus toxoid protein was consistently included and confirmed the specificity for the increase of IgG binding to hCoV S proteins in COVID-19 patient sera and in SARS-CoV-2 immunized macaque sera. Interestingly, we did observe a significant increase in antibody levels to tetanus toxoid following vaccination in humans, indicating also aspecific immune activation by the Pfizer-BioNTech mRNA vaccine, which has been observed by others as well (*van Gils et al., 2021*).

These responses after immunization with a vaccine containing a similar prefusion-stabilized SARS-CoV-2 S protein as many current COVID-19 vaccines demonstrate that these narrow focused vaccines might already induce low levels of cross-reactive antibodies. However, the strength of the response should be an important consideration as low levels of antibodies to viral antigens and/or the presence of predominantly non-neutralizing antibodies could potentially lead to antibody-dependent enhancement (ADE) of infection (*Wan et al., 2020*). Currently, ADE has not been reported widely for SARS-CoV-2 (*Arvin et al., 2020*) and for other hCoVs there is no consensus (*Lee et al., 2020*). In addition, it remains unclear in clinical studies if preexisting immunity against other hCoVs influences the severity of COVID-19 (*Westerhuis et al., 2020*; *Dugas et al., 2021*; *Anderson et al., 2021*; *Sagar et al., 2021*). In these 50 COVID-19 patients, we did not find evidence of a clear association between the presence of cross-reactive antibodies and COVID-19 disease severity.

As we show that most of the cross-reactivity is targeting the S2 subdomain of the S protein, we propose that this subdomain should receive more attention in vaccine design. The influence of the pre- or postfusion conformation of S2 on this level of cross-reactivity remains unknown as only a postfusion S2 construct is currently available. In addition, we can only speculate about the functionality of these S2 targeting antibodies as no neutralization assays for other hCoVs are performed in this study. However, others have described protective neutralizing antibodies targeting the S2 subdomain (*Huang et al., 2021*). Furthermore, analogous to some influenza virus hemagglutinin stem antibodies (*DiLillo et al., 2016*), antibody effector functions might also contribute to protective immunity by S2 antibodies (*Zohar and Alter, 2020*; *Shiakolas et al., 2020*). Additional possibilities to improve breadth include the use of consensus or mosaic vaccine designs similar to what has been done in HIV-1 vaccine research (*Sliepen et al., 2019*; *Corey and McElrath, 2010*; *Cohen et al., 2021*). In light of the ongoing COVID-19 pandemic, only a few S2 subdomain mutations are reported in SARS-CoV-2 variants (*Hodcroft, 2021*). Therefore, a S2-based booster vaccine after SARS-CoV-2 vaccination could be used to amplify and broaden the immune response.

The results from this study on the presence and specificity of cross-reactive antibodies to other hCoVs after SARS-CoV-2 infection and vaccination emphasize the feasibility of broad coronavirus vaccines and may guide future vaccine designs.

## Materials and methods

**Key resources table**

| Reagent type (species) or resource | Designation | Source or reference | Identifiers | Additional information |
|---|---|---|---|---|
| Gene (viral structural protein) | SARS-CoV-2 spike | GenBank | ID: MN908947.3 | N/A |
| Gene (viral structural protein) | SARS-CoV spike | GenBank | ID: ABD72984.1 | N/A |
| Gene (viral structural protein) | MERS-CoV spike | GenBank | ID: AHI48550.1 | N/A |
| Gene (viral structural protein) | hCoV-OC43 spike | GenBank | ID: AAT84362.1 | N/A |
| Gene (viral structural protein) | hCoV-HKU1 spike | GenBank | ID: Q0ZME7 | N/A |
| Gene (viral structural protein) | hCoV-229E spike | GenBank | ID: NP_073551.1 | N/A |
| Gene (viral structural protein) | hCoV-NL63 spike | GenBank | ID: AKT07952.1 | N/A |
| Cell line (human) | FreeStyle HEK293F cells | Thermo Fisher | Cat#: R79007; RRID:CVCL_D603 | N/A |
| Cell line (human) | HEK293T/ACE2 cells | Schmidt et al | Ref: 51 | N/A |
| Biological sample (human) | Human sera, post-infection | *Brouwer et al., 2020* | Ref: 52 | N/A |
| Biological sample (human) | Human sera, post-vaccination | Appelman et al. | Ref: 53 | N/A |
| Biological sample (cynomolgus macaque) | Cynomolgus macaque sera | *Brouwer et al., 2021* | Ref: 32 | N/A |
| Antibody | Goat-anti-human IgG-PE (goat polyclonal) | Southern Biotech | Cat#: 2040-09; RRID: AB_2795648 | 'Used at 1.3 µg/mL' |
| Peptide, recombinant protein | Prefusion-stabilized S protein ectodomain of SARS-CoV-2 | *Brouwer et al., 2020* | Ref: 52 | N/A |
| Peptide, recombinant protein | Prefusion-stabilized S protein ectodomain of SARS-CoV | . | Ref: 52 | N/A |
| Peptide, recombinant protein | SARS-CoV-2 nucleocapsid | Sanquin Research | Ref: 54 | Provided by Gestur Vidarsson and Federica Linty of Sanquin Research, Amsterdam, the Netherlands |
| Peptide, recombinant protein | tetanus toxoid | Creative Biolabs | Cat#: Vcar-Lsx003 | N/A |
| Peptide, recombinant protein | SARS-CoV-2 S1 subdomain | ABclonal Biotechnology | Cat#: RP01262 | N/A |
| Peptide, recombinant protein | SARS-CoV2 S2 subdomain | ABclonal Biotechnology | Cat#: RP01267 | N/A |
| Chemical compound, drug | 1-Ethyl-3-(3-dimethylaminopropyl) carbodiimide | Thermo Fisher Scientific | Cat#: A35391 | N/A |
| Chemical compound, drug | Sulfo-N-hydroxysulfosuccinimide | Thermo Fisher Scientific | Cat#: A39269 | N/A |
| Chemical compound, drug | Polyethylenimine hydrochloride (PEI) MAX | Polysciences | Cat#: 24765-1 | N/A |
| Software, algorithm | GraphPad Prism 8.3.0 | GraphPad | N/A | N/A |
| Software, algorithm | MATLAB 9.6 (R2019a) | MATLAB | N/A | N/A |
| Other | Luminex Magplex beads | Luminex | Cat#: MC10043-01 | N/A |
| Other | MAGPIX | Luminex | Cat#: MAGPIX-XPON4.1-RUO | N/A |
| Other | NiNTA agarose beads | QIAGEN | Cat#: R90115 | N/A |

| Reagent type (species) or resource | Designation | Source or reference | Identifiers | Additional information |
|---|---|---|---|---|
| Other | Superose6 increase 10/300 GL column | Cytiva | Cat#: 29091596 | N/A |
| Other | 4–12% NuPAGE Bis-Tris | Thermo Fisher | Cat#: NP0321BOX | N/A |
| Other | Novex colloidal blue staining kit | Invitrogen | Cat#: LC6025 | N/A |
| Other | HMW-Native Protein Mixture | GE Healthcare | Cat#: 17044501 | N/A |
| Other | Nano-Glo Luciferase Assay System | Promega | Cat#: N1130 | N/A |
| Other | GloMax | Turner BioSystems | Cat#: 9101-002 | N/A |

## Study population

Sera of 50 SARS-CoV-2-infected adults were collected 4–6 weeks after symptom onset through the cross-sectional COVID-19-specific antibodies (COSCA) study (NL 73281.018.20) as described previously (*Brouwer et al., 2020*). In short, all participants had at least one nasopharyngeal or oropharyngeal swab positive for SARS-CoV-2 as determined by qRT-PCR (Roche LightCycler480, targeting the Envelope-gene 113 bp), and 49 of 50 participants were serological positive for SARS-CoV-2 (Wantai, WS-1096, total SARS-CoV-2 RBD targeting antibodies). Patient demographics and medical history were collected and used to score the maximum disease severity by using the WHO disease severity criteria of COVID-19 (https://www.who.int/publications/i/item/clinical-management-of-covid-19, accessed on 16-10-2020). The COSCA study was conducted at the Amsterdam University Medical Centre, location AMC, the Netherlands, and approved by the local ethical committee (NL 73281.018.20). All individuals provided written informed consent before participating.

Sera of 30 healthy donors were kindly provided by the Dutch Institute for Public Health and Environment (RIVM). Sera donated in 2019 were used to exclude possible exposure to SARS-CoV-2. No seasonal influence was observed when analyzing IgG binding to the seasonal coronaviruses of control sera obtained at different calendar months (data not shown).

Pre- and post-vaccination sera of 45 SARS-CoV-2-naïve participants were collected as part of the prospective serological surveillance cohort (S3-study) conducted in two tertiary medical centers in the Netherlands (*Sikkens et al., 2021*). The study was approved by the local ethical committee of both hospitals, and all participants gave written informed consent before participating (NL73478.029.20). In short, participants were included since the beginning of the pandemic in the Netherlands, March 2020, and asked to visit the hospital every 1–2 months to test for a history of SARS-CoV-2 infection by determining their serological response using the Wantai ELISA and to fill in a questionnaire about COVID-19-related symptoms and prior SARS-CoV-2 qRT-PCR results. Only participants without a history of a SARS-CoV-2 infection were included in this current study. In January 2021, these SARS-CoV-2-naïve participants were vaccinated twice with the Pfizer-BioNTech mRNA vaccine. Serum samples were obtained approximately 21 days after the first and 28 days after the second vaccination. Additional information about the study population and the SARS-CoV-2-specific response following vaccination is described by *Appelman et al., 2021*.

## Cynomolgus macaques

Cynomolgus macaques (n = 6) received 50 µg of SARS-CoV-2 S-I53-50NP adjuvanted with 500 µg of MPLA liposomes (Polymun Scientific, Klosterneuburg, Austria) diluted in PBS by intramuscular route at weeks 0, 4, and 10. Blood samples were taken at baseline and weeks 2, 4, 6, 8, 10, and 12 after vaccination. Additional information is described by *Brouwer et al., 2021*.

## Cell lines

HEK293F cells were sourced from Invitrogen, and HEK293T cells expressing the SARS-CoV-2 receptor ACE2 were provided by Paul Bieniasz of The Rockefeller University and verified by FACS. Cells have been tested for *Mycoplasma* contamination.

## Protein designs

A prefusion-stabilized S protein ectodomain of SARS-CoV-2 and SARS-CoV with a T4 trimerization domain and hexahistidine (His) tag were previously described (*Brouwer et al., 2020*). A

prefusion-stabilized S protein ectodomains construct of hCoV-HKU1 was designed as described previously (*Kirchdoerfer et al., 2016*). Specifically, a gene encoding amino acids 1–1276, in which the furin cleavage site (amino acids 752–756) was replaced with a GGSGS sequence and amino acids at position 1067 and 1068 were mutated to prolines, was ordered and cloned into a pPPI4 plasmid containing a T4 trimerization domain followed by a hexahistidine tag. The same procedure was used to generate prefusion-stabilized S proteins of MERS-CoV, hCoV-229E, hCoV-NL63, and hCoV-OC43. For the prefusion-stabilized S protein ectodomain of hCoV-229E, a gene encoding amino acids 1–1082, with proline substitutions at positions 871 and 872, was ordered. The prefusion-stabilized S protein ectodomain construct of hCoV-NL63 consisted of amino acids 1–1263 and introduction of proline substitutions at positions 1052 and 1053. For MERS-CoV, a gene encoding amino acids 1–1214 with proline substitutions at positions 1060 and 1061 was obtained, and the furin cleavage site (amino acids 748–751) replaced with an ASVG sequence. The prefusion-stabilized S protein ectodomain construct of hCoV-OC43 was designed as described previously (*Tortorici et al., 2019*). Specifically, this construct consisted of amino acids 1–1263, which had the furin cleavage site replaced with a GGSGG sequence and proline substitutions at positions 1070 and 1071. GenBank IDs MN908947.3 (SARS-CoV-2), ABD72984.1 (SARS-CoV), AHI48550.1 (MERS-CoV), AAT84362.1 (hCoV-OC43), Q0ZME7 (hCoV-HKU1), NP_073551.1 (hCoV-229E), and AKT07952.1 (hCoV-NL63) served as templates for the protein designs.

## Protein expression and purification

The tetanus toxoid protein was acquired from Creative Biolabs. SARS-CoV-2 nucleocapsid protein (*Larsen et al., 2021*) was kindly provided by Gestur Vidarsson and Federica Linty of Sanquin Research, Amsterdam, the Netherlands. All other proteins were produced in HEK293F cells (Invitrogen) maintained in FreeStyle medium (Life Technologies). Transfections were performed using polyethylenimine hydrochloride (PEI) MAX (Polysciences) at 1 mg/L and the expression plasmids at 312.5 µg/L in a 3:1 ratio in 50 mL OptiMEM (Gibco) per liter. Supernatants were harvested 7 days post transfection by centrifugation at 4000 rpm for 30 min followed by filtration of the supernatant using 0.22 µM Steritop filter units (Merck Millipore). The His-tagged proteins were purified from the clarified supernatant with affinity chromatography using NiNTA agarose beads (QIAGEN). Eluates were concentrated and buffer exchanged to PBS using 100 kDa molecular weight cutoff (MWCO) Vivaspin centrifugal concentrators. Further purification to remove aggregated and monomeric protein fractions was performed using size-exclusion chromatography on a Superose6 increase 10/300 GL column (GE Healthcare) using PBS as buffer. Trimeric S proteins were eluted at a volume of approximately 13 mL. Fractions containing trimeric protein were pooled and concentrated using 100 kDa MWCO Vivaspin centrifugal concentrators. The resulting protein concentrations were determined using a Nanodrop 2000 Spectrophotometer, and proteins were stored at –80°C until needed. Information regarding protein integrity was obtained from Native PAGE analysis, which showed all proteins consisted of trimers (*Figure 2—figure supplement 5*). 4–12% NuPAGE Bis-Tris gels (Invitrogen) were used with a Novex colloidal blue staining kit (Invitrogen) and HMW-Native Protein Mixture (GE Healthcare) as a marker.

## Protein coupling to Luminex beads

Proteins were covalently coupled to Luminex Magplex beads using a two-step carbodiimide reaction and a ratio of 75 µg protein to 12.5 million beads for SARS-CoV-2 S protein. Other proteins were coupled equimolar to SARS-CoV-2 S protein. Luminex Magplex beads (Luminex) were washed with 100 mM monobasic sodium phosphate pH 6.2 and activated by addition of sulfo-N-hydroxysulfosuccinimide (Thermo Fisher Scientific) and 1-ethyl-3-(3-dimethylaminopropyl) carbodiimide (Thermo Fisher Scientific) and incubated for 30 min on a rotator at room temperature. The activated beads were washed three times with 50 mM MES pH 5.0, and the proteins were diluted in 50 mM MES pH 5.0 and added to the beads. The beads and proteins were incubated for 3 hr on a rotator at room temperature. Afterward, the beads were washed with PBS and blocked with PBS containing 2% BSA, 3% fetal calf serum, and 0.02% Tween-20 at pH 7.0 for 30 min on a rotator at room temperature. Finally, the beads were washed and stored at 4°C in PBS containing 0.05% sodium azide and used within 6 months. Detection of the His-tag on each S protein-coupled bead was used to confirm the amount of protein on the beads.

## Luminex assays

Optimization experiments determined the optimal concentration for detection of hCoV S protein binding antibodies to be 1:10,000 dilution for COVID-19 patients, 1:50,000 for SARS-CoV and SARS-CoV-2 detection in cynomolgus macaques, and 1:500 for detection of binding antibodies to other hCoV S proteins in cynomolgus macaques. 50 μL of a working bead mixture containing 20 beads per μL of each region was incubated overnight with 50 μL of diluted serum. Plates were sealed and incubated on a plate shaker overnight at 4°C. The next day, plates were washed with TBS containing 0.05% Tween-20 (TBST) using a hand-held magnetic separator. Beads were resuspended in 50 μL of goat-anti-human IgG-PE (Southern Biotech) and incubated on a plate shaker at room temperature for 2 hr. Afterward, the beads were washed with TBST and resuspended in 70 μL MAGPIX Drive Fluid (Luminex). The beads were agitated for a few minutes on a plate shaker at room temperature and then read-out was performed on a MAGPIX (Luminex). The resulting MFI values are the median of approximately 50 beads per well and were corrected by subtraction of MFI values from buffer and beads-only wells. A titration of serum of one convalescent COVID-19 patient as well as positive and negative controls was included on each plate to confirm assay performance. Assays were performed twice (technical replicates) with similar results (*Figure 2—figure supplement 6*).

## Depletion luminex assay

Convalescent COVID-19 patient sera were diluted 1:10,000 with addition of 10 μg/mL soluble recombinant SARS-CoV-2 prefusion-stabilized S protein, monomeric in a postfusion confirmation S1 or S2 subdomain (ABclonal Biotechnology) or without protein (as undepleted controls). Immunized macaque sera were diluted 1:500 with addition of 30 μg/mL SARS-CoV-2 S protein. Incubation was performed in uncoated 96-wells plates on a shaker at room temperature for 1 hr. Then, 50 μL of a working bead mixture containing 20 beads per μL of each region was added, and the plates were incubated 2 hr on a shaker at room temperature. After 2 hr, the above-described Luminex protocol was continued starting with the TBST washes and the goat-anti-human IgG-PE incubation.

## SARS-CoV-2 pseudovirus neutralization assay

The pseudovirus neutralization assay was performed as described previously (*Brouwer et al., 2021*). Briefly, HEK293T cells expressing the SARS-CoV-2 receptor ACE2 (*Schmidt et al., 2020*) were seeded in poly-L-lysine-coated 96-wells plates, and the next day triplicate serial dilutions of heat-inactivated serum samples were prepared, mixed 1:1 with SARS-CoV-2 pseudovirus, incubated for 1 hr at 37°C, and then added in a 1:1 ratio to the cells. After 48 hr, the cells were lysed, transferred to half-area 96-wells white microplates (Greiner Bio-One), and Luciferase activity was measured using the Nano-Glo Luciferase Assay System (Promega) with a GloMax system (Turner BioSystems). Relative luminescence units were normalized to the units from cells infected with pseudovirus in the absence of serum. Neutralization titers ($ID_{50}$) were the serum dilution at which infectivity was inhibited 50%.

## Statistical analysis

Luminex data were log transformed prior to any statistical analysis. p-Values below 0.05 were considered statistically significant. Mann–Whitney U tests for unpaired comparisons, Wilcoxon matched-pairs signed-rank test for paired comparisons, Friedman test with Dunn's multiple comparisons test for paired comparisons with correction for multiple testing, and Spearman's correlations were performed in GraphPad Prism 8.3.0. An overview of all statistical tests, exact p-values, and 95% confidence Intervals is provided in *Supplementary file 3*. Principal component analysis was performed in MATLAB 9.6 (R2019a).

## Acknowledgements

We are thankful to the participants of the COSCA and the S3-study for their contribution to this research. In addition, we thank the research team of the RECoVERD/VIS-cohort study for assisting with the recruitment and inclusion of participants. We thank Gestur Vidarsson and Federica Linty of Sanquin, Amsterdam, the Netherlands, for providing the SARS-CoV-2 nucleocapsid protein. We thank Mathieu AF Claireaux, Aafke Aartse, Ronald Derking, and Jonne L Snitselaar for providing assay controls and Aafke Aartse for manuscript editing.

This work was supported by a Netherlands Organization for Scientific Research (NWO) Vici grant no. 91818627 (to RWS) and a NWO ZonMw grant no. 10430022010023 (to MKB); by the Bill & Melinda Gates Foundation grants INV-002022, INV008818 (to RWS) and INV-024617 (to MJvG); by the Amsterdam UMC Corona Research Fund (to MKB); by an AMC Fellowship from Amsterdam UMC (to MJvG), and by an NHMRC career development fellowship (to AWC). The funders had no role in study design, data collection, data analysis, data interpretation, or data reporting.

## Additional information

### Group author details

**Amsterdam UMC COVID-19 S3/HCW study group**

**Diederik van de Beek**: Department of Neurology, Amsterdam Neuroscience, AMC, Amsterdam, Netherlands; **Justin de Brabander**: Center for Experimental and Molecular Medicine, Amsterdam Infection & Immunity, AMC, Amsterdam, Netherlands; **Matthijs C Brouwer**: Department of Neurology, Amsterdam Neuroscience, AMC, Amsterdam, Netherlands; **David TP Buis**: Department of Internal Medicine, Amsterdam Infection & Immunity, VUmc, Amsterdam, Netherlands; **Nora Chekrouni**: Department of Neurology, Amsterdam Neuroscience, AMC, Amsterdam, Netherlands; **Niels van Mourik**: Department of Intensive Care Medicine, Amsterdam Infection & Immunity, AMC, Amsterdam, Netherlands; **Sabine E Olie**: Department of Neurology, Amsterdam Neuroscience, AMC, Amsterdam, Netherlands; **Edgar JG Peters**: Department of Internal Medicine, Amsterdam Infection & Immunity, VUmc, Amsterdam, Netherlands; **Tom DY Reijnders**: Center for Experimental and Molecular Medicine, Amsterdam Infection & Immunity, AMC, Amsterdam, Netherlands; **Michiel Schinkel**: Center for Experimental and Molecular Medicine, Amsterdam Infection & Immunity, AMC, Amsterdam, Netherlands; **Alex R Schuurman**: Center for Experimental and Molecular Medicine, Amsterdam Infection & Immunity, AMC, Amsterdam, Netherlands; **Marleen A Slim**: Department of Intensive Care Medicine, Amsterdam Infection & Immunity, AMC, Amsterdam, Netherlands; **Yvo M Smulders**: Department of Internal Medicine, Amsterdam Infection & Immunity, VUmc, Amsterdam, Netherlands; **Alexander PJ Vlaar**: Department of Intensive Care Medicine, Amsterdam Infection & Immunity, AMC, Amsterdam, Netherlands; **W Joost Wiersinga**: Center for Experimental and Molecular Medicine, Amsterdam Infection & Immunity, AMC, Amsterdam, Netherlands

### Funding

| Funder | Grant reference number | Author |
|---|---|---|
| Nederlandse Organisatie voor Wetenschappelijk Onderzoek | 91818627 | Rogier W Sanders |
| ZonMw | 10430022010023 | Marije K Bomers |
| Bill and Melinda Gates Foundation | INV-002022 | Rogier W Sanders |
| Bill and Melinda Gates Foundation | INV008818 | Rogier W Sanders |
| Bill and Melinda Gates Foundation | INV-024617 | Marit J van Gils |
| Amsterdam UMC | Corona Research Fund | Marije K Bomers |
| Amsterdam UMC | AMC Fellowship | Marit J van Gils |
| National Health and Medical Research Council | Career development fellowship | Amy W Chung |

The funders had no role in study design, data collection and interpretation, or the decision to submit the work for publication.

## Author contributions
Marloes Grobben, Conceptualization, Data curation, Formal analysis, Investigation, Methodology, Project administration, Validation, Visualization, Writing – original draft, Writing – review and editing; Karlijn van der Straten, Conceptualization, Data curation, Formal analysis, Investigation, Methodology, Project administration, Resources, Validation, Visualization, Writing – original draft, Writing – review and editing; Philip JM Brouwer, Mitch Brinkkemper, Pauline Maisonnasse, Nathalie Dereuddre-Bosquet, Brent Appelman, AH Ayesha Lavell, Lonneke A van Vught, Dirk Eggink, Tom PL Bijl, Hugo DG van Willigen, Elke Wynberg, Bas J Verkaik, Orlane JA Figaroa, Peter J de Vries, Tessel M Boertien, Jonne J Sikkens, Roger Le Grand, Menno D de Jong, Maria Prins, Resources, Writing – review and editing; Judith A Burger, Meliawati Poniman, Melissa Oomen, Investigation, Writing – review and editing; Amsterdam UMC COVID-19 S3/HCW study group, Resources; Marije K Bomers, Funding acquisition, Resources, Writing – review and editing; Amy W Chung, Conceptualization, Funding acquisition, Methodology, Writing – review and editing; Godelieve J de Bree, Resources, Supervision, Writing – review and editing; Rogier W Sanders, Marit J van Gils, Conceptualization, Funding acquisition, Methodology, Resources, Supervision, Writing – review and editing

## Author ORCIDs
Marloes Grobben (iD) http://orcid.org/0000-0002-1559-9592
Karlijn van der Straten (iD) http://orcid.org/0000-0001-6373-9302
Philip JM Brouwer (iD) http://orcid.org/0000-0002-2902-7739
Marit J van Gils (iD) http://orcid.org/0000-0003-3422-8161

## Ethics
Human subjects: For the convalescent COVID-19 sera, approval was granted by the ethical review board of the Amsterdam University Medical Centers, location AMC, the Netherlands (NL 73281.018.20). For the healthy controls, anonymized leftover serum from routine diagnostics for which ethical approval was waived by the ethics committee of the National Institute of Public Health and the Environment was used. The study was performed in accordance with the guidelines for sharing of patient data of observational scientific research in emergency situations as issued by the Commission on Codes of Conduct of the Foundation Federation of Dutch Medical Scientific Societies (https://www.federa.org/federa-english). For the pre- and post-vaccination sera, approval was granted by the local ethical committees of both Amsterdam University Medical Centers, location AMC and location VUMC (NL73478.029.20). All animals from which samples were used in this study were housed in IDMIT infrastructure facilities (CEA, Fontenay-aux-roses) under BSL-2 and BSL-3 containment when necessary (animal facility authorization #D92-032-02, Préfecture des Hauts de Seine, France) and incompliance with European Directive 2010/63/EU, the French regulations and the Standards for Human Care and Use of Laboratory Animals, of the Office for Laboratory Animal Welfare (OLAW, assurance number #A5826-01, US). The protocols were approved by the institutional ethical committee 'Comité d'Ethique en Expérimentation Animale du Commissariat; l'Energie Atomique et aux Energies Alternatives' (CEtEA #44) under statement number A20-011. The study was authorized by the 'Research, Innovation and Education Ministry' under registration number APAFIS#24434-2020030216532863v1.

## Decision letter and Author response
Decision letter https://doi.org/10.7554/eLife.70330.sa1
Author response https://doi.org/10.7554/eLife.70330.sa2

---

# Additional files

## Supplementary files
• Supplementary file 1. Sociodemographics, clinical characteristics, and severity scoring for COVID-19 patients.

• Supplementary file 2. Sequence identity matrices for the ectodomains of all hCoV S proteins. Sequence identity matrices were composed of all coronavirus spike proteins in this study. All sequences comprise only the truncated ectodomain of each spike as was used to generate the recombinant proteins. S1, S2, and RBD were defined as noted in the corresponding GenBank sequences (see Materials and methods). Multiple sequence alignments were performed and sequence identities calculated using Clustal Omega 1.2.4.

- Supplementary file 3. Overview of statistical tests, exact p-values, and 95% confidence intervals.
- Transparent reporting form
- Source data 1. Source data of all panels of all figure supplements.

### Data availability

All relevant data is included in the paper and supplementary materials.

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
