## [Editor Report]

To develop a pan-coronavirus vaccine, the authors investigated the presence and specificity of cross-reactive antibodies against the spike proteins of human coronaviruses after SARS-CoV-2 infection and vaccination and quantified cross-reactive antibodies.

---

## [Decision Letter]

**Decision letter after peer review:**

Thank you for submitting your article "Cross-reactive antibodies after SARS-CoV-2 infection and vaccination" for consideration by *eLife*. Your article has been reviewed by 2 peer reviewers, one of whom is a member of our Board of Reviewing Editors, and the evaluation has been overseen by Sara Sawyer as the Senior Editor. The following individual involved in review of your submission has agreed to reveal their identity: Andrew T McGuire (Reviewer #3).

Essential revisions:

In this study, Grobben et al. examines whether binding antibodies that cross-react with the spikes of diverse coronaviruses are elicited by SARS-CoV-2 infection. The manuscript is well written, and the figures are laid out in an easy to interpret manner. This study will be of interest to those who are interested in developing pan coronavirus vaccines. However, the following issues should be addressed before acceptance:

1. What are the relative contributions to the SARS-CoV-2 neutralizing activity in the sera upon depletion of S1 or S2 subdomains? It would be very interesting to assess whether there a reduction of serum neutralizing activity following S2 mAb depletion as would imply that the cross binding S2 antibodies might have neutralizing activity against diverse CoVs.

2. Do the authors have access to human post-vaccine sera? It would be interesting to compare the results of natural infection to vaccination.

3. Could any of the differences in cross reactive binding be due to differences in the stability of the various spike proteins? Presumably all of the spike-ectodomain designs will stabilize the proteins in the pre-fusion state, but it is not demonstrated that they are. Misfolding could lead to the destruction of relevant conformation-dependent cross reactive epitopes.

4. Related to the point 4 above, is the S2 used in the depletion studies in the pre-fusion or post-fusion conformation? Since the 2 conformations are quite different structurally, it's likely that they are antigenically distinct, and may therefore fail to deplete out relevant cross-reactive serum antibodies.

If neither of the points (3 and 4) can be easily addressed experimentally these caveats should be pointed out in the discussion.

5. The authors suggest that the macaques are likely CoV naïve, however the data suggest that this may not be true. If the animals are truly naive, then all of the cross-reactive antibodies should have been elicited by the SARS-CoV-2 spike protein and similarly depleted by the SARS-CoV-2 spike protein. For example, in Figure 4b all of the SARS-CoV-2 spike binding antibodies are depleted by the SARS-CoV-2 spike, but only about 70% of the SARS and OC43 binding antibodies were depleted. Do the authors have an alternative explanation for this?

*Reviewer #1:*

In this study, Grobben et al. evaluated the cross-reactivity of anti-SARS-CoV-2 S antibodies against the S proteins of a variety of coronaviruses. The results shown by the authors are important and should be published for the post-vaccination era. However, the cross-reactivity is mostly evaluated by only the binding affinity to the S proteins.

The results shown by Grobben et al. are basically important and should be published for the post-vaccination era. However, the cross-reactivity is mostly evaluated by only the binding affinity to the S proteins. The authors should evaluate the neutralizing activity of these antibodies using live viruses and/or pseudoviruses. Using all coronaviruses might be difficult, but some (e.g., seasonal coronaviruses such as 229E and NL63) should be used.

Evaluation of neutralizing activity would be important for the authors' conclusion. Evaluating the binding affinity to a variety of coronavirus S protein is very important and insightful. However, to consider its biological activity, neutralizing effect is much more important, because binding affinity and neutralizing affinity are not necessarily correlated.

*Reviewer #3:*

Here the authors were seeking to assess whether SARS-CoV-2 infection elicited antibodies that cross react with diverse human CoVs using a Luminex platform. The major conclusions are that cross-binding antibodies are elicited by SARS-CoV-2 infection, and that there is a positive association between the magnitude of the response and the sequence similarity to SARS-CoV-2. As a result, most cross-reactive antibodies target the S2 region which is the most conserved among diverse CoVs. The study also shows that immunization with a SARS-CoV-2 spike derived nanoparticle vaccine similarly elicits cross reactive CoV antibodies in non-human primates. Overall, the data suggests that the S2 domain might be important for the development of pan CoV vaccines.

The study is largely observational, and conclusions are supported by the data. The major limitation is that it is not clear what role the measured cross-reactive antibodies might play in protective immunity. If a function could be assigned to the cross-reactive antibodies ie neutralization or ADCC, then these findings might be more impactful for guiding the design of pan CoV vaccines.

What are the relative contributions to the SARS-CoV-2 neutralizing activity in the sera upon depletion of S1 or S2 subdomains? It would be very interesting to assess whether there a reduction of serum neutralizing activity following S2 mAb depletion as would imply that the cross binding S2 antibodies might have neutralizing activity against diverse CoVs.

Do the authors have access to human post-vaccine sera? It would be interesting to compare the results of natural infection to vaccination.

Could any of the differences in cross reactive binding be due to differences in the stability of the various spike proteins? Presumably all of the spike-ectodomain designs will stabilize the proteins in the pre-fusion state, but it is not demonstrated that they are. Misfolding could lead to the destruction of relevant conformation-dependent cross reactive epitopes.

Related to the above point, is the S2 used in the depletion studies in the pre-fusion or post-fusion conformation? Since the 2 conformations are quite different structurally, it's likely that they are antigenically distinct, and may therefore fail to deplete out relevant cross-reactive serum antibodies.

If neither of the points can be easily addressed experimentally these caveats should be pointed out in the discussion.

The authors suggest that the macaques are likely CoV naïve, however the data suggest that this may not be true. If the animals are truly naive, then all of the cross-reactive antibodies should have been elicited by the SARS-CoV-2 spike protein and similarly depleted by the SARS-CoV-2 spike protein. For example, in Figure 4b all of the SARS-CoV-2 spike binding antibodies are depleted by the SARS-CoV-2 spike, but only about 70% of the SARS and OC43 binding antibodies were depleted. Do the authors have an alternative explanation for this?

---

## [Author Response]

Essential revisions:In this study, Grobben et al. examines whether binding antibodies that cross-react with the spikes of diverse coronaviruses are elicited by SARS-CoV-2 infection. The manuscript is well written, and the figures are laid out in an easy to interpret manner. This study will be of interest to those who are interested in developing pan coronavirus vaccines. However, the following issues should be addressed before acceptance:1. What are the relative contributions to the SARS-CoV-2 neutralizing activity in the sera upon depletion of S1 or S2 subdomains? It would be very interesting to assess whether there a reduction of serum neutralizing activity following S2 mAb depletion as would imply that the cross binding S2 antibodies might have neutralizing activity against diverse CoVs.

Indeed it would be very interesting to assess neutralizing activity against diverse CoVs, as noted by both reviewers. However, assays to determine neutralizing activity to the seasonal human coronaviruses are quite complicated and therefore can currently not be performed at our laboratory. It could be possible to determine SARS-CoV-2 neutralizing activity after S1 or S2 subdomain depletion as suggested, however we do not think this would improve general knowledge on function of S2 directed cross-reactive antibodies. This neutralizing potential has been shown in better ways by others as we noted in our discussion. To better explain our limitation and the presence of knowledge on the functionality of S2 antibodies by others, we further elaborated on this in our discussion (lines 361-366).

2. Do the authors have access to human post-vaccine sera? It would be interesting to compare the results of natural infection to vaccination.

We thank the reviewer for this interesting suggestion. Hence, we sought a collaboration to investigate the humoral cross-reactivity following Pfizer-BioNTech mRNA vaccination in 45 SARS-CoV-2 naïve individuals and found a similar increase in hCoV antibodies. We incorporated these results in our manuscript (Figure 5 and Figure 5—figure supplement 1, and lines 295-311).

3. Could any of the differences in cross reactive binding be due to differences in the stability of the various spike proteins? Presumably all of the spike-ectodomain designs will stabilize the proteins in the pre-fusion state, but it is not demonstrated that they are. Misfolding could lead to the destruction of relevant conformation-dependent cross reactive epitopes.

We agree that it would be possible that differences in stability and confirmation may affect our results. We have used the tools available to us to characterize the produced proteins, however due to lack of well-described antibodies we cannot confirm integrity of specific epitopes. We did confirm that all spike proteins formed trimers by BN-PAGE gel and acknowledge that this was missing information in our methods section. To give more insight on our proteins we have added this information (lines 468-472) and the corresponding gel pictures (Figure 2—figure supplement 5). In addition, we noted conformational differences as a limitation in our Discussion section (lines 340-343).

4. Related to the point 4 above, is the S2 used in the depletion studies in the pre-fusion or post-fusion conformation? Since the 2 conformations are quite different structurally, it's likely that they are antigenically distinct, and may therefore fail to deplete out relevant cross-reactive serum antibodies.

The S2 subdomain used in depletion studies was a commercially available reagent consisting of monomeric S2 in the post-fusion state. To the best of our knowledge, no pre-fusion S2 construct has been described yet and therefore this was not available to us. We agree that it would be interesting to repeat the experiment when such a protein would become available as indeed it might further amplify the depletion that we observed. We have added more details about the S2 in both the methods section (line 507-508) and the Results section (line 210). And noted lack of a pre-fusion S2 as a limitation in the discussion (lines 361-363).

If neither of the points (3 and 4) can be easily addressed experimentally these caveats should be pointed out in the discussion.

We have addressed these points experimentally as much as possible and have included the caveats of this study more clearly in the discussion.

5. The authors suggest that the macaques are likely CoV naïve, however the data suggest that this may not be true. If the animals are truly naive, then all of the cross-reactive antibodies should have been elicited by the SARS-CoV-2 spike protein and similarly depleted by the SARS-CoV-2 spike protein. For example, in Figure 4b all of the SARS-CoV-2 spike binding antibodies are depleted by the SARS-CoV-2 spike, but only about 70% of the SARS and OC43 binding antibodies were depleted. Do the authors have an alternative explanation for this?

The reviewer raises an important point and we have studied our results in more detail again. Our depletion assay was able to remove a large portion of the antibodies binding to other human CoV spikes. Since the antibody levels for macaques were low, the signal-to-noise ratio in our assay was not very high providing lower sensitivity and in addition some variability was observed. Together, this may be part of the explanation why the depletion rates do not reach 100% in all cases. We cannot fully exclude the possibility that pre-existing immunity in macaques contributed to this incomplete depletion. However, hCoV infections are not frequently described in macaques and SARS-CoV is not a common circulating hCoV. Therefore, it is more likely that this incomplete deletion is associated with assay variability when signal-to-noise ratio are limited (for the alpha hCoVs) and possible conformational differences between the spike proteins (for SARS-CoV). We have noted this in the discussion (lines 340-343). Additionally, we do agree that solely noting that the macaques are assumed to be naïve is not well enough supported by our data and therefore we adapted the discussion to better explain the limitation in our study that the animals were not tested for possible previous coronavirus infections prior to the immunizations (lines 333-336).

We repeated our depletion experiment to confirm the incomplete depletion observed for SARS-CoV. We again concluded that we could not fully deplete this serological response. As described above, we think this may be explained by conformation changes between the spike proteins used for immunization and depletion.